# An Integrated Capture of Red Mud and One-Step Heat-Treatment Process to Recover Platinum Group Metals and Prepare Glass-Ceramics from Spent Auto-Catalysts

**Chuan Liu [1,2], Shuchen Sun [1,2,\*], Ganfeng Tu [1,2] and Faxin Xiao [1,2]**

[1] School of Metallurgy, Northeastern University, Shenyang 110819, China; neuliuchuan@outlook.com (C.L.); tugf@smm.neu.edu.cn (G.T.); xiaofx@smm.neu.edu.cn (F.X.)
[2] Key Laboratory for Recycling of Nonferrous Metal Resources, Shenyang 110819, China
[\*] Correspondence: sunsc@smm.neu.edu.cn; Tel.: +86-024-8368-9195

**Abstract:** Co-treatment for two kinds of hazardous solid waste is an effective method to reduce cost and increase recycle efficiency of value resource. This work developed an integrated process based on capture of red mud (RM) and a one-step heat-treatment process to efficiently recover PGMs from spent auto-catalysts (SAC) and reuse RM simultaneously. Firstly, the iron oxide in RM was reduced to metallic iron to capture PGMs by the reduction process, without the addition of an extra reducing agent, since SAC contained abundant organic volatiles. Then, the mixed waste of SAC and RM was melted under high temperature with additives of CaO and $H_3BO_3$. More than 99% of PGMs can be extracted under the optimal conditions of 40–50 wt% of RM addition, 14 wt% of $H_3BO_3$ addition, 0.7–0.8 of basicity, 1500 °C of temperature, and 40 min of holding time. In addition, PGM content in obtained glassy slag was less than 1 g/t. The mechanism of iron trapping PGMs was also discussed in detailed, which mainly contained two stages: migration of PGMs and separation of PGM-bearing alloy and slag phases. Besides, the obtained glassy slag was further prepared into glass-ceramic by a one-step heat-treatment process. It was found that the prepared glass-ceramic has good thermostability and an excellent stabilizing effect on heavy metals. Overall, the results indicated that the developed integrated smelting–collection process is an efficient and promising method for the reutilization of SAC and RM.

**Keywords:** recycling; platinum group metals; capture of red mud; glass-ceramic; spent automotive catalyst

## 1. Introduction

A substantial increase in the use of automotives has resulted in a great deal of exhaust emissions, which contain numerous toxic exhaust gases including carbon monoxide (CO), nitrogen oxide ($NO_x$), and hydrocarbons (HCs), causing a serious threat to the environment. With the promulgation of strict laws and regulations worldwide on automotive emission standards, more than 85% of the vehicles install ternary catalysts to convert exhaust gases into less harmful products because of the large specific surface and small gas flow resistance. The auto-catalyst (AC) employs platinum group metals (PGMs: platinum (Pt), palladium (Pd), rhodium (Rh)) as the active components due to their high catalytic activity, high-temperature stability, and selection. Generally, Pt and/or Pd are the active catalytic elements for CO and HC oxidation, while Rh is the active catalytic element for $NO_x$ reduction [1,2]. The global consumption of PGMs is huge in catalytic conversion due to the rapid development of the automotive industry. Approximately 45% of Pt, 65% of Pd, and 84% of Rh were used in the automotive industry in 2018 [3]. Typically, the AC is composed of an outer steel shell, a fiber blanket, and a PGM-bearing monolithic. At present, monolithic materials are mainly composed of iron/magnesium ceramic cordierite ($2FeO/2MgO \cdot 2Al_2O_3 \cdot 5SiO_2$), into which 90% $\gamma$-$Al_2O_3$ and other metal oxide additives,

such as $CeO_2$ and $ZrO_2$, are coated on the inner surface. After long-term use, the spent auto-catalyst (SAC), as one of the fastest-growing hazardous solid wastes, has been extensively studied in recent years [2,4]. The concentrations of PGMs in SACs, approximately 0.1–0.2%, are much higher than those in natural PGM resources, less than 0.001% [5,6]. Furthermore, SACs generally contain heavy metals such as lead (Pb), chromium (Cr), vanadium (V), and nickel (Ni), flame retardants, and burnable organic constituents, which may cause ground and water pollution. As a result, the SAC has been classified as hazardous solid waste and has restricted disposal in landfills [7]. Therefore, it is necessary to recycle PGMs from SACs in terms of economic and environmental protection. This not only reduces the amount of PGMs extracted from ores and simultaneously disposes of hazardous waste to generate substantial environmental benefits, but also recovers PGMs to bring about considerable economic value.

There are two major types of recycling techniques reported: pyrometallurgical techniques and hydrometallurgical techniques [8–10]. Hydrometallurgical methods have been conventionally used to recycle SACs under strong acid solution (sulfuric acid, $H_2SO_4$; hydrochloric acid, HCl; aqua regia) and alkaline solution (sodium hydroxide, NaOH; NaCN). Although hydrometallurgical processes have many advances such as low-energy consumption, selectivity of chemical agents on metals, and easy operation, they have several apparent disadvantages. For instance, this process inevitably requires abundant chemical reagents and produces large amounts of wastewater and residues, thereby causing secondary pollution [10]. Moreover, hydrometallurgy is not suitable for low concentrations of PGMs in SACs. Furthermore, this process has an unstable PGM recovery rate, especially for Rh (<90%) [11].

Pyrometallurgical methods have improved from the perspectives of pollution control along with process efficiency in recent years. This technology, which is simplified and environmentally friendly, has attracted extensive attention in the application of recovering PGMs from SACs [12–14]. In particular, the metal collection process of base metals such as copper (Cu) and iron (Fe) is a common technology for the enrichment of PGMs with low concentrations and has been widely applied in many corporations with advanced metallurgical and refining technologies. In this process, crushed SAC is usually mixed with a collector of base metals, fluxes ($SiO_2$, CaO, or $Al_2O_3$), and carbon reductants and then smelted under a high temperature. After smelting, the PGMs are concentrated and enriched in the alloy phase, and carriers and fluxes are transferred into the slag phase with a low viscosity. The alloy phase is readily separated from slag due to the large different density of the slag and metallic phase, resulting in the PGM-containing alloy sinking into the bottom of the furnace [15,16]. Ding et al. [17] used an iron melting collection process to recover PGMs from SACs with a mixed reductant, iron collector, and fluxes. The results showed that over 99% of PGMs were recovered under the optimal conditions, namely, $CaO/Na_2O = 35:20$, $CaF_2$ 5 wt%, $Na_2B_4O_7$ 8.5 wt%, Fe 15 wt%, and C 5 wt%. Morcali [18] proposed the capture of PGMs by an iron matte at approximately 1000 °C and found that 99% of Pt, 99% of Pd, and 97% of Rh were extracted under optimal conditions with three flux additions, namely, a mass ratio of $B_2O_3/Na_2O$ of 0.72 and 10 g of $FeS_2$. However, the methods mentioned above invariably need the addition of a large amount of collector and fluxes. Therefore, reducing the amount of added collector and fluxes is one of the main methods to reduce the cost during the metal collection process.

Red mud (RM), an industrial byproduct, is generated during the Bayer process in an aluminum electrolytic plant. The current annual production of RM reaches approximately 120 million tons [19]. RM is considered hazardous solid waste due to its high alkalinity (pH 10–13), which results in intensive environmental and economic issues worldwide [20]. To meet strict environmental laws and regulations, numerous investigations have been conducted to treat RM, including direct utilization and reutilization. For example, RM is directly used as an absorbent for a pollutant, neutralizing reagent of acid wastes, and construction materials. Nevertheless, these methods typically cause new environmental risks, and a minor fraction of RM can be accommodated [21]. RM is generally composed

of 4.52–50.6 wt% Fe, 4.42–16.06 wt% $Al_2O_3$, 0.98–5.34 wt% $TiO_2$, and a small number of rare elements. Therefore, many studies on the recovery of valuable metals from RM have been carried out, including physical beneficiation, pyrometallurgical processes, hydrometallurgical processes, and hydro-pyrometallurgical methods. Among these processes, pyrometallurgical technology is an effective process to recover Fe from RM via a reduction–smelting process, which attracts extensive attention [22,23]. In our previous work [24], we proposed the smelting–collection of Pt from SACs via cooperation with different carriers along with the addition of an iron collector. The Pt in SACs with different carriers was effectively captured by metallic iron at 1550 °C for 120 min. Meanwhile, glassy slag, environmentally friendly slag, was also obtained, in which the Pt content was only 9.4 g/t. Kim et al. [25] adopted the reduction–smelting process to recover PGMs from SACs and spent mobile phone printed circuit boards (PCBs) with the addition of waste copper slag. In this process, the waste copper slag was used not only as flux material but also as the collector of PGMs. Moreover, the plastic component from PCB served as a reducing agent of iron oxides in waste copper slag. The results indicated that more than 95% of PGMs can be recovered under a high temperature of over 1400 °C. Nevertheless, few works have focused on the recovery of PGMs from SACs using other hazardous solid waste as the flux and collector. Although many techniques have been carried out to recycle valuable metals from SACs and RM, an approach that comprehensively recovers PGMs from SACs and Fe from RM via a cooperative smelting–collection process without the addition of collectors and reducing agents has not yet been developed. Typically, the iron oxides and oxides contained in RM can act as collectors of PGMs and fluxing materials of slag, respectively. Meanwhile, a large number of organic volatiles in RM can also be reducing agents to reduce iron oxides into metallic iron.

In the present study, a novel process utilizing RM as the flux material and PGM collector for the reduction–collection smelting process was attempted. Moreover, the combination of SACs and RM will improve the flexibility of the operative feed material in recycling process. The PGM and Fe recovery from the mixture waste of SACs and RM associated with slag chemistry was studied via high-temperature smelting experiments. Furthermore, the knowledge on the reduction–collection smelting process during the cooperative treatment was improved using analytical data and thermodynamic calculations.

## 2. Materials and Methods

### 2.1. Starting Materials

The SAC and RM in this study were collected from Jiangsu and Shandong provinces, respectively. These wastes were predried at 105 °C for 12 h for chemical analysis. The chemical compositions of the SAC and RM were determined by X-ray fluorescence (XRF, ZSX PrimusII, Rigaku Corporation, Tokyo, Japan) spectroscopy and the content of PGMs was analyzed by inductively coupled plasma mass spectrometry (ICP-MS, EXPEEC700, Focused Photonics Inc., Beijing, China), as shown in Table 1. $SiO_2$, $Al_2O_3$, and MgO were the major oxides of the SAC, and the PGM content of the SAC was 857 g/t. $SiO_2$, $Al_2O_3$, $Fe_tO$, and $TiO_2$ were the main oxides of RM. X-ray diffraction (XRD, D8 Advance, Bruker AXS Co., Ltd., Karlsruhe, Germany) patterns of the SAC and RM are presented in Figure 1. The major phase of the SAC is cordierite ($2MgO \cdot 2Al_2O_3 \cdot 5SiO_2$), and the RM is mainly composed of goethite, hematite, boehmite, and polygorskite. Notably, the PGM-bearing phase cannot be determined due to the limitation of XRD. Previous studies have indicated that some PGMs will be encapsulated into the inner layer of the carrier after long-term use. However, the PGMs in SACs mostly remain in the metallic state due to their extremely strong stability [26]. Analytical-grade CaO and $H_3BO_3$ were introduced as fluxes to control slag compositions, which were produced by Sinopharm Chemical Reagent Co., Ltd., Beijing, China.

**Table 1.** Chemical compositions of the SAC and RM used for the experiments.

| Components | Concentration (wt%) | |
| --- | --- | --- |
| | **SAC** | **RM** |
| $SiO_2$ | 34.6 | 7.18 |
| $Al_2O_3$ | 31.0 | 16.36 |
| CaO | 1.7 | 1.05 |
| MgO | 8.9 | 0.14 |
| TFe | 2.5 | 38.74 |
| $K_2O$ | 0.3 | 0.18 |
| $Na_2O$ | | 2.88 |
| $ZrO_2$ | 4.1 | 0.21 |
| MnO | 0.6 | |
| S | | 0.004 |
| P | | 0.11 |
| V | | 0.15 |
| ZnO | 0.2 | |
| $Cr_2O_3$ | | 0.15 |
| $TiO_2$ | 0.7 | 6.22 |
| Pt/Pd/Rh [a] | 97.0/642.0/118.0 | |
| LOI [b] | 12.6 | 9.4 |

TFe: Total Fe; [a]: g/t; [b]: Loss mass in ignition.

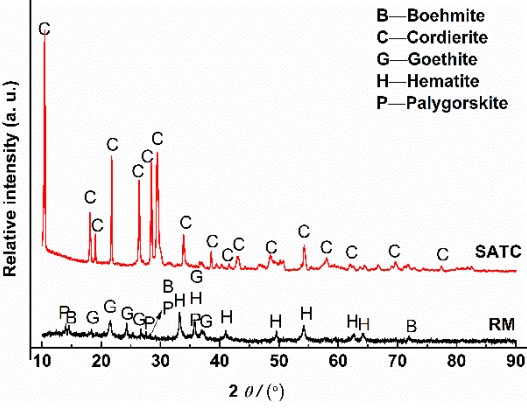

**Figure 1.** XRD patterns of SAC and RM.

Different mass ratios between the SAC and RM were selected to investigate the influence of RM addition on PGM recovery. Mixing mass ratios (SAC:RM) of 8:2, 7:3, 6:4, and 5:5 were selected to simplify the raw waste material-mixture preparation.

### 2.2. Experimental Procedure

The process flowsheet is shown in Figure 2. The high-temperature experiments were carried out in a medium-frequency inductive furnace (XZ 25, Shanghai Xiangda Co., Ltd., Shanghai, China). A 100 g measure of mixed waste material of SAC and RM (RM addition range of 20–50 wt%) was first mixed with CaO (13–21 wt%: 13–21 g), and 14 wt% $H_3BO_3$, and then thoroughly homogenized. The crucible containing homogenized mixed materials was transferred into the furnace and operated at a voltage of 200 V and current of 20 A (1200 °C) for 10 min to reduce iron oxides in RM by organic volatiles. Subsequently, the voltage and current were increased to 250 V and 25 A (1500 °C) for 40 min to ensure that the mixed material was fully melted. Then, the melt was rapidly poured into a circular truncated cone-type preheated mold. The slag and Fe-based alloy were readily separated due to the large difference in density between the slag and alloy. The Fe button was cleaned, weighed, and crushed, and the slag sample was finely ground into powder for subsequent chemical analysis. The compositions of the slag and alloy were measured using XRF. The

PGM content in the slag and alloy was determined by using ICP-MS. The PGM recovery was calculated by the following Equation (1):

$$R = (1 - \frac{M_1 \times C_1}{M_0 \times C_0}) \times 100\% \tag{1}$$

where $R$ is the PGM recovery, %; $M_0$ is the mass of the SAC; $C_0$ is the PGM content in the SAC; $M_1$ is the mass of the slag; and $C_1$ is the PGM content in the slag.

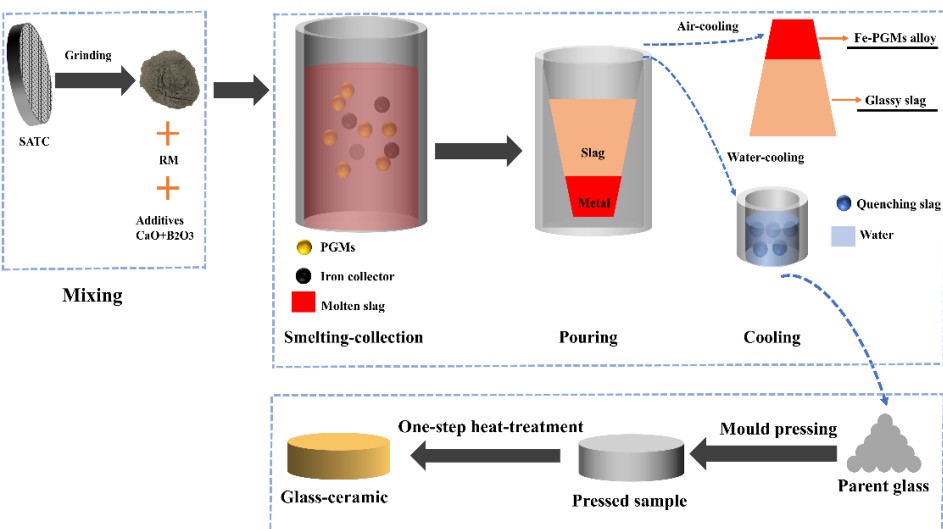

**Figure 2.** The simple flow diagram of the proposed technology in this study.

### 2.3. Analytical Methods

After the smelting process, the obtained PGM-bearing alloy and slag were weighed. The slag sample was finely ground to power for chemical analysis. The PGM-bearing alloy was crushed for the following assay. The compositions of the slag and alloy were determined by XRF; therein, the PGM content in the slag and alloy was analyzed by using ICP-MS. The crystal structure of the slag and PGM-bearing alloy was identified by XRD. Microstructure characterization of the obtained slag and alloy as well as prepared glass-ceramic were investigated using scanning electron microscopy (SEM, Ultra Plus, Zeiss, Heidenheim, Germany) with an accelerating voltage of 15 kV, equipped with an energy-dispersive X-ray spectroscopy (EDS, X-Max 50, Oxford Instruments, Oxford, UK) detector. In addition, the obtained glass slag and sintered glass-ceramic were subjected to toxicity characteristic leaching procedure (TCLP) experiments. The TCLP experiments were carried out under the guidance of Erol et al.'s [26] and Lv et al.'s works [27].

### 3. Result and Discussion

### 3.1. Optimization of the Cooperative Smelting Process

### 3.1.1. Effect of RM Addition on the PGM Recovery

RM was incorporated into the SAC as a flux material and collector. Smooth gravitation separation between the slag and PGM-bearing alloy was attained via the molten slag system that was assisted by RM. As shown in Table 1 and Figure 3a obtained by the thermodynamic calculation software of FactSage 7.2 (version 7.2, Thermfact/CRCT and GTT-Technologies, Montreal, Canada and Ahern, Germany), the RM was an excellent flux agent for the SAC smelting–collection process since it contains $SiO_2$, $Al_2O_3$, $TiO_2$, and $Fe_tO$. A previous study found that a multicomponent slag with more than five different oxides generally complicates slag design and hinders a clear expectation of the physicochemical properties of slag [28]. Restricting the number of oxides can effectively facilitate slag design for the smelting process. Nevertheless, the addition of RM did not considerably change the types of oxides in the SAC, and thereby, the slag system can be expected by designing

a mixture of SAC and RM. Furthermore, there was another advantage for the use of RM as an additive. This is mainly because of the Fe present in Fe$_t$O in the RM. A high total Fe content (approximately 38.74 wt%) can provide a great deal of metallic Fe through the carbothermal reduction process, which served as the collector of PGMs. As a result, the addition of an extra collector was avoided. Therefore, the addition of RM not only can reduce the amount of flux addition but also avoid the addition of collectors, simultaneously increasing the types of hazardous solid waste treatment.

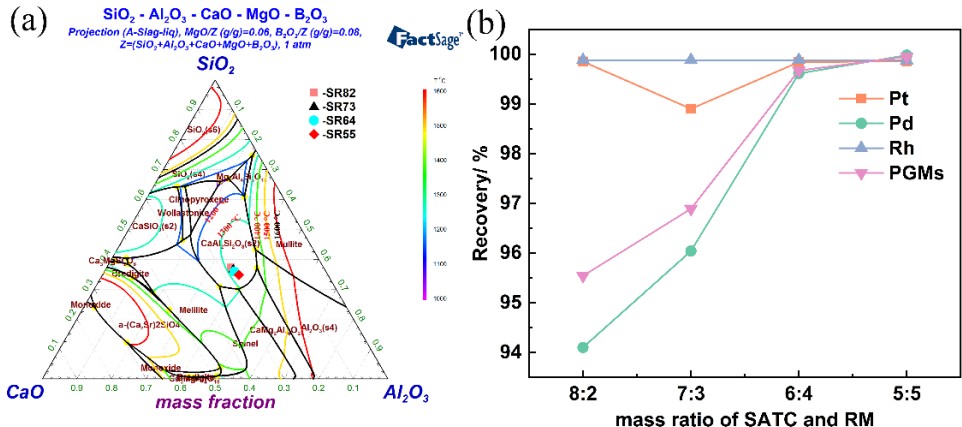

**Figure 3.** Melting point of mixed wastes of SAC and RM in SiO$_2$-Al$_2$O$_3$-CaO-6 wt%MgO—8 wt%B$_2$O$_3$ (**a**) and effect of RM addition on PGM recovery (**b**) (H$_3$BO$_3$ addition of 14 wt%, basicity of 0.8, smelting temperature of 1500 °C, and holding time of 40 min).

The smelting–collection experiment was carried out in a reducing atmosphere due to the presence of organic volatiles in RM that reduced Fe$_t$O into metallic Fe. The effect of the amount of RM addition on the PGM recovery rate was studied in the addition range from 20 to 50 wt%, basicity (mass ratio of CaO to SiO$_2$) of 0.8, H$_3$BO$_3$ addition of 14 wt%, temperature of 1500 °C, and holding time of 40 min. The relationship between the recovery of PGMs and RM addition is presented in Figure 3b. As shown in Figure 3b, the recovery of Pd first significantly increased with the increasing RM addition, and then the rate of increase in the recovery decreased and finally reached a maximum value. For instance, the recovery of Pd increased from 94.1% at RM addition of 20 wt% to 99.61% at that of 40 wt%, increasing by 5.51%, while the recovery of Pd increased only by 0.37% as the addition of RM continuously increased to 50 wt%. This may be attributed to the different concentrations of PGMs in the SAC, resulting in the different PGM recoveries under the same smelting conditions. As seen from Table 1, the concentrations of Pt, Pd, and Rh in the SAC were 97, 642, and 118 g/t, respectively, and thus Pt and Rh were readily and completely captured by the collector. As a result, the recoveries of Pt and Rh, with lower concentrations of PGM, were higher than that of Pd when the amount of collector was low because of the small addition of RM. Moreover, the Pt and Rh recoveries remained relatively stable, approximately 100%, regardless of the amount of RM added. This is mainly caused by the comprehensive effects of RM. Regarding this study, the more RM added, the higher the amount of Fe in the mixed waste of SAC and RM and the lower concentration of PGMs. Therefore, on the one hand, the metallic PGM particles are more readily collected by Fe collector to further form larger alloy droplets since the RM addition increased, and thus, the PGM-containing alloy droplets more easily collided with each other, gathered, grew up, and further settled to the bottom of the crucible. On the other hand, the PGMs were more widely dispersed in mixed waste of SAC and RM when the RM addition increased, and hence the PGMs were more difficult to trap. These two factors led to the increase in and stability of PGM recoveries, which showed an opposite effect on the PGM recovery. In the process of RM addition, a balance effect that increased the amount of collector of Fe and decreased the PGM concentration in mixed waste caused by addition of RM could

be achieved. This effect was predominant at 40–50 wt% of RM addition, resulting in a maximum PGM recovery approaching 100%. Therefore, comprehensively considering the PGM recoveries and amount of slag, it is reasonable to summarize that mixed waste of SAC and RM with 40–50 wt% RM additions were feasible.

### 3.1.2. Effect of Basicity on PGM Recovery

The effect of basicity on PGM recovery was investigated in the range from 0.5 to 0.8 under the conditions that the RM addition, smelting temperature, and holding time were 40 wt%, 1500 °C, and 40 min, respectively. The basicity was determined according to the pseudo-ternary phase diagram drawn by FactSage 7.2, as shown in Figure 4a. The relationship between PGM recoveries and basicity is shown in Figure 4b. As indicated by Figure 4b, as the basicity increased, the comprehensive recovery of PGMs first increased and then reached approximately 100%, up to 99.91%. Therein, the recoveries of Pt were not significantly changed as basicity increased, increasing from 99.84% at a basicity of 0.5 to 99.85% at that of 0.8, only increased by 0.01%. Meanwhile, the Rh recovery increased from 99.2% at a basicity of 0.5 to 99.87% at that of 0.8, an increase of only 0.67%. Nevertheless, the Pd recovery gradually increased with the increase in basicity, increasing from 97% at 0.5 to 99.92% at 0.8 and increasing by 2.92%. These results show that the Pt, Pd, and Rh recovery change tendency is similar to the effect of RM addition on the PGM recoveries, which is caused by their relative content in the mixed waste of SAC and RM. Moreover, in this smelting system, the viscosity of slag decreased with the increase in basicity, as shown in Figure 4c, thereby affecting the recovery of PGMs. The viscosity of slag significantly affected the aggregation and sedimentation of alloy particles, and separation between the alloy and slag phases. Therefore, the viscosity of slag decreased gradually with the increase in basicity, resulting in the fluidity of slag being better, which contributed to the capture process. Hence, the optimum basicity for the recovery process was determined to be 0.7–0.8 since the comprehensive PGM recovery was more than 99% under these conditions.

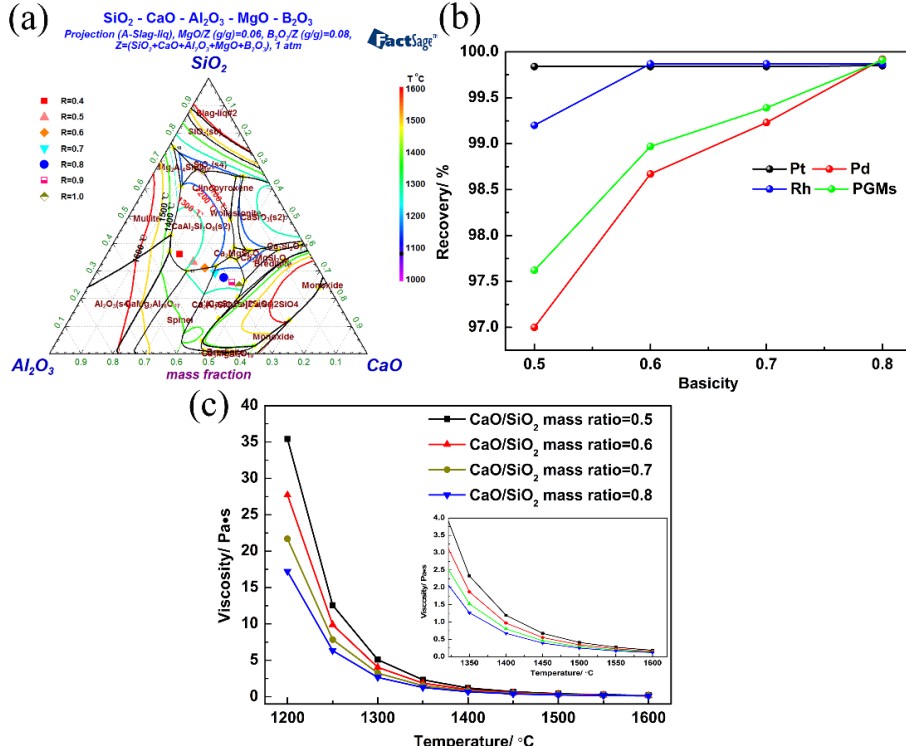

**Figure 4.** The phase diagram of mixed wastes SAC and RM under different basicity: (**a**) effect of basicity on the PGM recovery (**b**) (RM addition of 40 wt%, H$_3$BO$_3$ addition of 14 wt%, smelting temperature of 1500 °C, and holding time of 40 min), and curve of viscosity and temperature for different basicity (**c**).

Finally, the optimal smelting parameters were selected as 40–50 wt% of additions of RM, and 0.7–0.8 of basicity when the smelting temperature and holding time was 1500 °C, and 40 min, respectively, under which the recoveries of Pt, Pd, Rh, and PGMs reached 99.85%, 99.92%, 99.87%, and 99.91%, respectively.

### 3.2. Confirmation Experiments

The confirmation experiment (CE) was conducted under the obtained optimum conditions above. These optimal conditions were used to recover PGMs from 500 g mixed waste of SAC and RM. After the smelting–collection process, 62.13 g PGM-enriched metal was obtained, and the concentrations of Pt and Rh in obtained slag were below the detection limit, while that of Pd was only 0.86 g/t. The recoveries of Pt and Rh were nearly 100%, and the Pd and comprehensive PGMs were 99.93% and 99.93%, respectively. The compositions of the slag and metal phases are shown in Table 2.

**Table 2.** The compositions of slag and metal phases generated from the confirmation experiment.

| Products | Compositions (wt%) | | | | | | | | | |
|---|---|---|---|---|---|---|---|---|---|---|
| Slag phase | $SiO_2$ 30.26 | CaO 18.84 | $Al_2O_3$ 30.68 | MgO 6.78 | $B_2O_3$ 8.32 | TFe 0.75 | $TiO_2$ 2.82 | Pt * BDL | Pd * 0.86 | Rh * BDL |
| Metal phase | TFe 88.2 | Si 1.79 | C 3.9 | Mn 0.62 | Pt * 392.6 | Pd * 2945.3 | Rh * 459.2 | | | |

BDL: Below detection limit value; *: g/t.

### 3.2.1. Analysis of the Slag Phase

The chemical reaction and species changes of the slag phase during the co-treatment process were analyzed using the Equilib module in FactSage 7.2 [29]. The equilibrium calculation results are presented in Figure 5a, and was carried out by adding mixed wastes and fluxing materials at a basicity of 0.8 when the RM addition was 40 wt%. As indicated by Figure 5a, when the temperature ranged from 1000 to 1294 °C, four insoluble phases, anorthite ($CaAl_2Si_2O_8$), spinel ($MgAl_2O_4$), $CaTiO_3$, and olivine ($[Mg,Fe]_2SiO_4$), were included, while these insoluble phases continuously decreased with increase in the temperature and gradually decomposed into liquid slag. These decomposition reactions of insoluble phases are presented in Equations (2)–(5). Moreover, it is also indicated in Figure 5a that $B_2O_3$ has all been melted into the molten slag before 1000 °C. Therefore, combined with the viscosity curve, from Figure 4c, and the multi-phase reactions, it is feasible to select the smelting temperature of 1500 °C. The phase of the obtained slag is presented in Figure 5b, which reflects that the slag revealed a completely amorphous phase, indicating that the glassy slag was successfully prepared by this proposed technology. In addition, the leaching characteristic tests of heavy metals including Zn, Cr, Mn, and Pb in slag were carried out to verify whether the leaching efficiency of heavy metals met the limit value, as shown in Table 3. The leaching efficiencies of all heavy metals were below the restriction standard of the US-EPA.

$$CaAl_2Si_2O_8 = CaO + Al_2O_3 + 2SiO_2 \tag{2}$$

$$MgAl_2O_4 = MgO + Al_2O_3 \tag{3}$$

$$CaTiO_3 = CaO + TiO_2 \tag{4}$$

$$[Mg,Fe]_2SiO_4 = 2MgO + 2FeO + SiO_2 \tag{5}$$

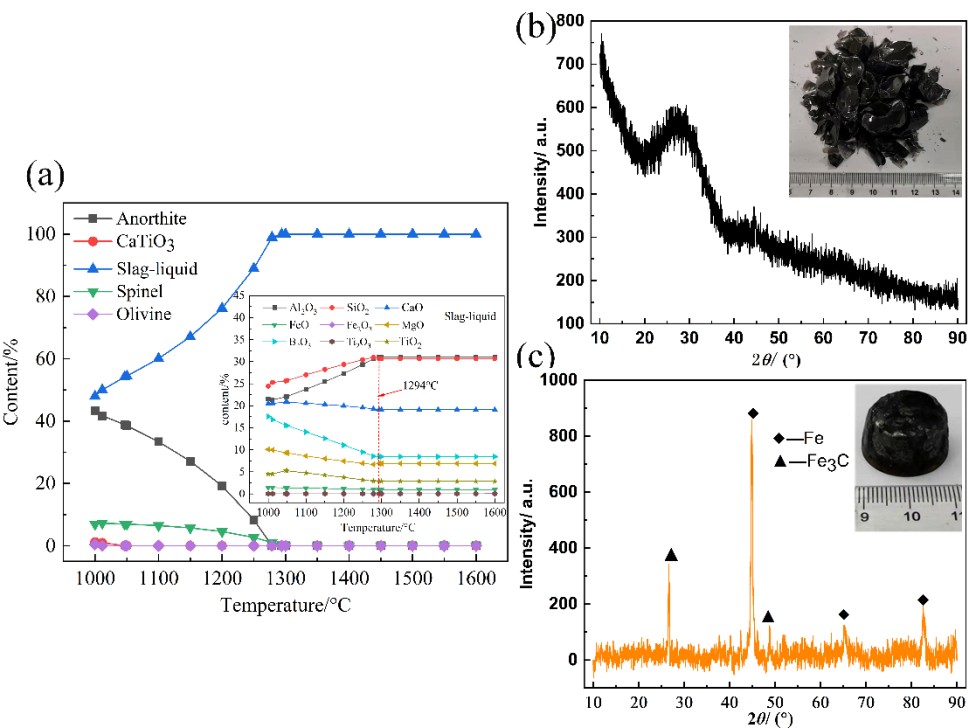

**Figure 5.** Simulation of the reaction equilibrium (**a**) and XRD results of the obtained slag (**b**) and PGM-bearing alloy (**c**) for confirmation experiment.

**Table 3.** TCLP results of produced glassy slag sample.

| Samples | TCLP Results (mg/L) | | | |
|---|---|---|---|---|
| | **Zn** | **Cr** | **Mn** | **Pb** |
| Glassy slag | 1 | 0.072 | 4.7 | BDL |
| GC0.5 | 0.34 | BDL | 3.22 | BDL |
| GC1 | 0.33 | BDL | 3.69 | BDL |
| GC2 | 0.29 | BDL | 3.77 | BDL |
| GC3 | 0.28 | BDL | 2.44 | BDL |
| GC4 | 0.28 | BDL | 2.48 | BDL |
| US-EPA limit | 500 | 5 | 5 | 5 |

BDL: Below detection limit value. GC0.5: The glass-ceramic obtained by heat-treatment for 0.5 h.

### 3.2.2. Analysis of the PGM-Enriched Alloy

The PGM-bearing Fe alloy was formed by a combination of PGMs in the SAC and Fe in RM in the smelting–collection process. The chemical compositions of the obtained PGM-bearing alloy are listed in Table 2. As indicated by Table 2, the PGM-bearing alloy was mainly composed of Fe and small amounts of Si, C, and Mn, in which the PGM content was over 0.3 wt%. To investigate the formation of the metal phase, XRD was introduced, and the results are shown in Figure 5c. The XRD patterns showed that the metal phase was mainly composed of a metallic iron phase (PDF#96-901-3415), and a small amount of iron carbide (PDF#96-101-0932). Due to the low concentration of PGMs in the obtained alloy, the corresponding PGM-bearing iron alloys were not detected by XRD. To further analyze the morphology and microstructure of PGM-bearing alloy, images of the formed alloy were observed by SEM, as shown in Figure 6a. It can be found that this alloy was primarily divided into two phases, namely, dark and embedded gray areas. Furthermore, the component analysis of the two areas is shown in Figure 6b,c by EDS analysis. Point 1 in the dark matrix was mainly composed of Fe and Pd, namely, the Fe–Pd phase, while point 2 in the gray region mainly contained Fe and small amounts of Pt, Pd, and Rh. Specifically, during the process of point scanning, the results might be inaccurate, as the

adjacent area could be contained. Besides, the mapping profiles of Fe, Pt, Pd, Rh, Si, and Mn are presented in Figure 6d–i by EDS analysis. As indicated by the results, Pd and Rh were evenly distributed in the Fe matrix, while Pt only existed in certain regions contained in the Fe matrix. Combined with the results of points and mapping scanning, it can be concluded that the obtained alloy was mainly composed of an Fe–Pd–Rh matrix phase and an embedded Pt phase. Meanwhile, small amounts of Si and Mn were detected, which were generated from the smelting–collection process and doped in the alloy phase as impurities.

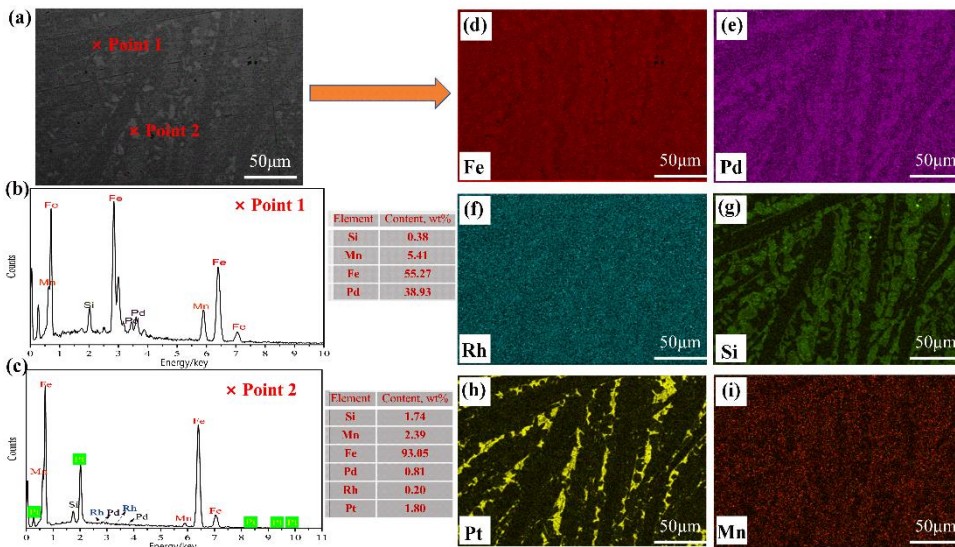

**Figure 6.** The profile of obtained PGM-bearing alloys observed via SEM (**a**), EDS component analysis at different regions for point 1 and point 2 (**b**,**c**), mapping images of Fe, Pt, Pd, Rh, Si, and Mn (**d**–**i**).

### 3.2.3. Economic Evaluation

The economic evaluation was also assessed to verify the prospect of the proposed method in this study. The cost of the confirmation experiment was simply calculated by this proposed method and a conventional technology that is smelted by the direct addition of fluxing materials [16]. Especially, in the calculation process, the energy consumption and labor costs were not considered due to the uncertainty in industrial production. The calculation results are listed in Table 4. As shown in Table 4, the total cost of disposing of 1 kg of SAC is USD 0.1994 by using the proposed method which is lower than that of SAC by using a traditional method, which is approximately 0.0964 USD/kg. Meanwhile, this proposed method effectively can not only decrease the types and amount of fluxing materials addition, but also increase the types of hazardous solid waste treatment simultaneously. Therefore, this implied that this proposed cooperative smelting–collection process with RM is a technically and economically good prospect for recycling PGMs from SACs.

**Table 4.** The comparison of cost of disposing 1 kg of SAC.

| Reference Method [16] | | | This Work | | |
|---|---|---|---|---|---|
| Chemicals | Amount/kg | Cost/USD | Chemicals | Amount/kg | Cost/USD |
| $SiO_2$ | 0 | 0 | $SiO_2$ | 0 | 0 |
| CaO | 0.3605 | 0.0649 | CaO | 0.2598 | 0.0468 |
| Iron powder | 0.1500 | 0.0825 | Iron powder | 0 | 0 |
| C | 0.0500 | 0.0080 | C | 0 | 0 |
| $Na_2B_4O_7$ | 0.0850 | 0.0374 | $H_3BO_3$ | 0.1400 | 0.1526 |
| $Na_2CO_3$ | 0.3498 | 0.0826 | RM | 0.5000 | 0 |
| $CaF_2$ | 0.0500 | 0.0204 | / | / | / |
| Sum | 1.0453 | 0.2958 | Sum | 0.8998 | 0.1994 |

### 3.3. Mechanism of Recovering PGMs via the Iron Capture Process

On the basis of the discussion above, the PGM-containing alloy can be formed in the process of recycling PGMs from SACs via a cooperative process with RM. Currently, there is no agreement on the mechanism of the capture process of iron for PGMs in the process of recycling PGMs from SAC. According to the analysis above, the mechanism of iron trapping was analyzed, as shown in Figure 7. It can be classified into two sections, namely, the migration of PGMs during the capture process of iron and the separation between alloy and slag during the formation of PGM-bearing alloys.

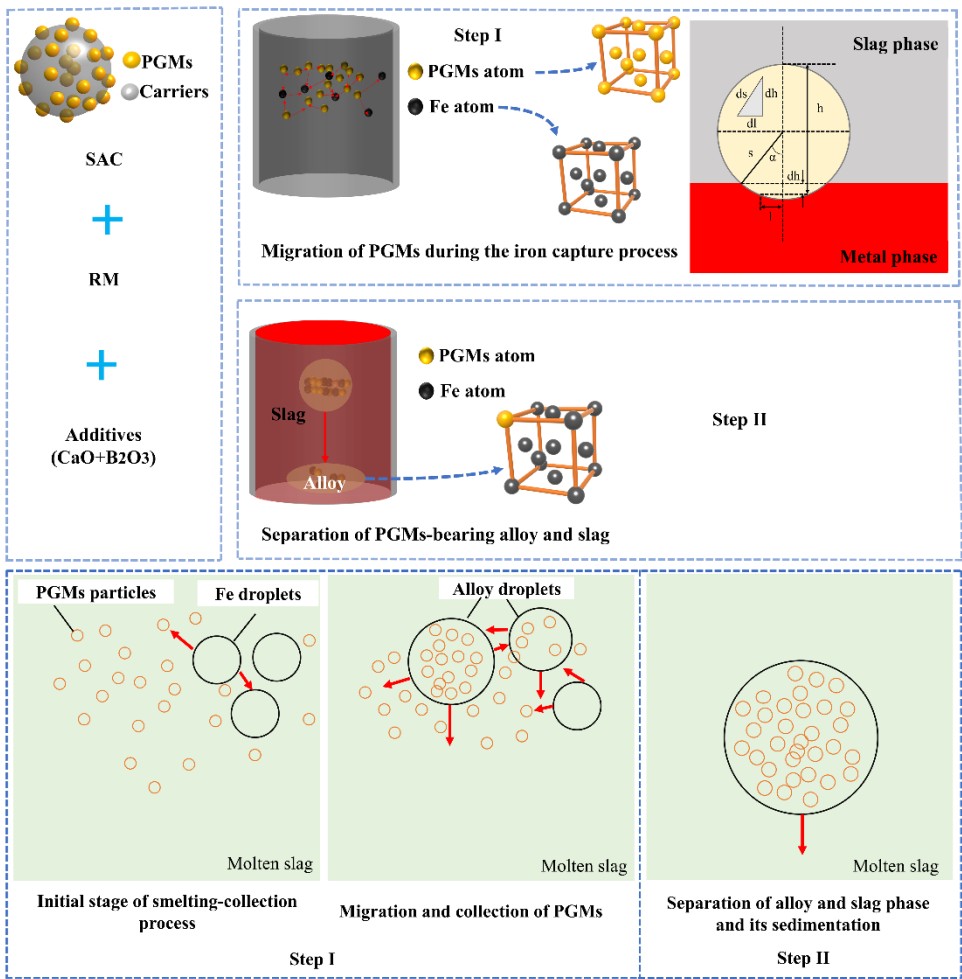

**Figure 7.** Mechanism of the proposed capture process of reduced iron from RM for PGMs contained in SAC.

### 3.3.1. Migration Process of PGMs

In the migration process of PGMs, the SAC, including carrier and PGMs, iron collector, and added fluxing materials were all melted as the smelting temperature increased. In the molten state, the PGMs encapsulated in the carrier were released into the molten slag system. Based on the application of chemical potential in multiphase equilibrium, the PGM particles suspended in the slag phase of $dn_{PGMs}^{slag}$ began to transfer into the iron collector metal phase (metal phase) under isothermal and isostatic pressure conditions, resulting in PGMs existing in the metal phase ($dn_{PGMs}^{metal}$) [15,30,31]. The Gibbs free energy of the system ($\triangle G$) can be expressed by Equation (6):

$$\triangle G = \triangle G^{metal} + \triangle G^{metal} = \mu_{PGMs}^{metal} \times dn_{PGMs}^{metal} + \mu_{PGMs}^{slag} \times dn_{PGMs}^{slag} \qquad (6)$$

where $\mu_{\text{PGMs}}^{\text{metal}}$ and $\mu_{\text{PGMs}}^{\text{slag}}$ are the chemical potentials of PGMs in the metal and slag phases, respectively.

$dn_{\text{PGMs}}^{\text{metal}} = -dn_{\text{PGMs}}^{\text{slag}}$ since the PGMs in the metals phase are generated from slag phase, and thereby, Equation (6) can be rewritten as Equation (7). When the transformation of PGMs from the slag phase to the metal phase is a spontaneous process, $\triangle G < 0$ must be met. Furthermore, $dn_{\text{PGMs}}^{\text{metal}} > 0$, namely, the metal phase contained PGMs, and thus, $\mu_{\text{PGMs}}^{\text{metal}} - \mu_{\text{PGMs}}^{\text{slag}} < 0$ in the migration process of PGMs from the slag to the metal phase.

$$\triangle G = \left( \mu_{\text{PGMs}}^{\text{metal}} - \mu_{\text{PGMs}}^{\text{slag}} \right) \times dn_{\text{PGMs}}^{\text{metal}} \tag{7}$$

Besides, the $\mu_{\text{PGMs}}^{\text{metal}}$ and $\mu_{\text{PGMs}}^{\text{slag}}$ are also regarded as the surface energy of PGMs in the metal and slag phases, respectively, which are affected by the surface tension of PGMs interacting with the slag and metal phases. To investigate the behavior of PGM migration caused by surface tension at different phase, considering the PGM particles as spherical inclusions of radius R [32], their position at a certain time in the system is shown in Figure 7. With the prolonged capture process, when the spherical inclusion transferred from slag to the metal phase in the form of infinitesimal contact with $\triangle$h, as shown in Figure 7, the interfacial tension between PGMs and slag, namely, $\sigma_{\text{PGMs−slag}}$, which was $2\pi l ds$, and that between the slag and metal phases, namely, $\sigma_{\text{slag−metal}}$, which was $2\pi l ds$, disappeared, while that between PGMs and the metal phase, namely, $\sigma_{\text{PGMs−metal}}$, which was $2\pi l ds$, appeared. Therefore, the change in the free energy of the system ($\triangle E$) can be calculated by Equation (8) during the transfer process of PGM inclusions from the slag phase to the metal phase:

$$\triangle E = \sigma_{\text{PGMs−metal}} \times 2\pi l ds - \sigma_{\text{PGMs−slag}} \times 2\pi l ds - \sigma_{\text{slag−metal}} \times 2\pi l dl \tag{8}$$

where $dl = ds \times \cos\alpha$, $0° < \alpha < 180°$.

Thus, Equation (8) can be expressed by Equation (9). If the PGM inclusions entered spontaneously into the metal phase from the slag phase, $\triangle E < 0$ must be satisfied, that is, $\sigma_{\text{PGMs−metal}} - \sigma_{\text{PGMs−slag}} - \sigma_{\text{slag−metal}} \times \cos\alpha < 0$. Zhang et al. and Yuan et al. [15,33] both pointed out that the surface tension between the metallic and nonmetallic phases is much larger than that of between the metallic phases, and hence, Equation (9) is always less than 0, which meets the thermodynamic condition of a spontaneous process. As a result, the PGMs transferred from slag phase to metal phase can be proceeded spontaneously. Besides, Chen [34] proposed the theory of delocalized electronics to further illustrate the spontaneous process for PGMs entering the metal phase from the slag phase. This theory revealed that slag was primarily composed of a complex network structure with silicate, of which there were localized electrons, while the metal atoms, including PGMs and iron collectors, were delocalized electrons. Therefore, the PGMs in slag readily transfer from the slag phase to the metal phase.

$$\triangle E = \sigma_{\text{PGMs−metal}} - \sigma_{\text{PGMs−slag}} - \sigma_{\text{slag−metal}} \times \cos\alpha \tag{9}$$

PGMs and iron can form continuous solid solutions under high-temperature processes [35]. The binding energies between PGMs and iron, based on the two cases of solid solution of interstitial solid solution and substitutional solid solution, were calculated via the first-principles density functional theory (DFT) method [35]. In the interstitial solid solution, the gap of the iron supercell was doped by PGM atoms, while the substitutional solid solution was regarded as an iron atom in the iron supercell and was replaced by PGMs. The results indicated that the PGMs existed in the iron matrix in the form of substitutional solid solution, and the corresponding binding energies of Fe–Pt, Fe–Pd, and Fe–Rh were −4.763 eV, −4.611 eV, and −4.752 eV, respectively. The formation energies of PGMs–Fe were all lower than 0, which indicated that the alloying process was spontaneous. Moreover, the formation energy of Fe–Pt was less than that of Fe–Pd and Fe–Rh, which can

further explain why the distribution of Pt in the iron matrix was different from that of Pd and Rh, as shown in Figure 6e,f,h. That is, Pt and Fe is more readily formed alloys than Pd and Rh.

### 3.3.2. Separation of Slag and Metal Phase

Generally, the sedimentation of alloy droplets is closely related to alloy droplet size, slag viscosity, density difference between slag and alloy, and collection time [36]. Benson et al. [37] found that the PGM particles in slag settled by their own gravity to the crucible bottom during the smelting process, and their particle diameter should be at least 200 μm. Typically, the PGM particle sizes in SACs range from 1 to 20 μm, less than 200 μm; therefore, PGM droplets cannot settle to the crucible bottom by their own gravity. Clearly, the addition of an iron collector can significantly increase the particle size of alloy droplets containing PGMs by forming larger metallic droplets during the recovery process. Takashi and Katsunori [38] found that the motion of alloy droplets in molten slag can be considered as sedimentation in the vertical direction during the smelting–collection process. The function between the velocity of sedimentation of the alloy droplet and these factors can be expressed by the Stokes formula, as shown in Equation (10)

$$v = \frac{g d^2 (\rho_A - \rho_s)}{18\eta} \tag{10}$$

where $v$ is the sedimentation velocity of the alloy, g is the acceleration of gravity, $d$ is the alloy droplet diameter, $\rho_A$ is the density of the alloy droplet, $\rho_s$ is the slag density, and $\eta$ is the slag viscosity.

Regarding this work, the PGM content in PGM-bearing alloy is less than 1 wt%, as shown in Table 2, and hence the alloy density in this study was approximately that of pure iron at a certain temperature. The function relationship between the density of pure iron ($\rho_{Fe}$) and temperature can be calculated by Equation (11) [37]. Therefore, the alloy density, in this CE, was approximately 7.07 g/cm$^3$ at 1500 °C, the smelting–collection temperature in this study, namely, $\rho_A$ = 7.07 g/cm$^3$. Furthermore, the density of slag ($\rho_s$) can be approximately calculated based on the temperature and compositions of molten slag, as shown in Equations (12) and (13) [39]:

$$\rho_{Fe} = 8.58 - \frac{0.853T}{1000} \tag{11}$$

where $T$ is the temperature, K.

$$V = \sum X_i V_i \tag{12}$$

$$\rho_s = \frac{M}{V} \tag{13}$$

where $X_i$ and $V_i$ (cm$^3$/mol) are the molar fraction and molar volume of i specie in the slag, respectively, and $V$ (cm$^3$/mol) and $M$ (g/mol) are the molar volume and molar mass of all oxides in the slag, respectively.

In addition, the molar volume ($V_i$) of oxide i can be obtained according to Equation (14), and the data of the molar volume of oxides in the obtained slag from the CE are presented in Table S1. Combined with the compositions of slag, in Tables 2 and S1, and Equations (12)–(14), $\rho_s$= 2.57 g/cm$^3$, and thereby the density difference between the metal and slag phases was up to 4.5 g/cm$^3$, which was much larger than the required minimum of 1.5 g/cm$^3$ [40]. Thus, the density difference between the metal and slag in this study was conducive to separating the alloy phase from the molten slag phase:

$$V_i = V_{i,1773K} - \frac{V_{i,1773K} \times (T - 1773)}{10,000} \tag{14}$$

where $V_{i,1773K}$ is the molar volume of oxide i at 1773 K, and $T$ is absolute temperature, K.

Slag viscosity plays a significant role in the separation process of slag and alloys, which is mainly affected by the compositions of slag and temperature. In this work, we introduced FactSage 7.2 to calculate the viscosity of slag obtained from the CE under different temperatures, and the calculation results are shown in Table S2. The slag viscosity was only 0.3 Pa·s at 1500 °C, namely, $\eta_{1500°C}$= 0.3 Pa·s, which was beneficial for alloy droplet collision, aggregation, and growth with each other, and enhanced the separation of the alloy from the slag phase. Therefore, the velocity of the alloy droplet was determined by the diameter of the alloy droplet when the densities of the slag and alloy, and the slag viscosity, were constant.

Therefore, PGMs contained in SAC can be effectively extracted by the iron smelting–collection process to form PGM-bearing alloys.

### 3.4. Synthesis of Glass-Ceramics

In the cooperative smelting–collection process with RM, the obtained slag, namely, glass slag, presented an amorphous phase, as shown in Figure 5b, which was mainly composed of $SiO_2$, $Al_2O_3$, CaO, $B_2O_3$, MgO, and $TiO_2$, as shown in Table 2. Therefore, this obtained glass slag can be taken as aluminoborosilicate glass, due to the high contents of $SiO_2$, $Al_2O_3$, and, $B_2O_3$, which is widely applied in various fields such as ion exchange materials, fiberglass, and metal seals because of its excellent mechanical properties, good chemical resistance, and strong thermal resistance [41]. These oxides in the obtained glass slag, $SiO_2$ and $Al_2O_3$, are the network formers, while $B_2O_3$ and $TiO_2$ are good nucleating agents [42]. Alkali metal ions such as $Ca^{2+}$ and $Mg^{2+}$, in general, are known as good glass network modifiers. While glass and glass-ceramics both exhibit good chemical stability, glass-ceramics have more advantages in high strength and wear resistance compared to glasses. As a result, glass-ceramics are more suitable for industrial applications of electronics, energy, and nuclear devices than glasses. Glass-ceramics are generally taken as a non-hazardous material, and thereby the preparation of glass-ceramics was applied to treat solid waste because this method has advantages such as reducing the volume of solid waste and achieving its reutilization [30]. The one-step heat-treatment process, which conducts the growth of nucleation and crystals at the same heat-treatment temperature, is a promising way to prepare glass-ceramics, and hence, it has received great attention in the last few years. Compared to the traditional heat-treatment process, namely, the two-step process (including nucleation stage and crystal stage), it has significant advantages of lower energy consumption, reduced environmental risks, and simplified process flow [43]. The crystallization temperature ($T_c$) was determined by simultaneous DSC-TGA (TA, SDTQ600), as shown in Figure 8a. Based on the results of DSC analysis, the heat-treatment process was conducted in a muffle furnace at a rate of 10 °C/min to 950 °C for 30 min to 240 min. The XRD phases of the prepared glass-ceramics under different heat-treatment times are presented in Figure 8b. As indicated by Figure 8b, the crystal of $CaAl_2Si_2O_8$ were extracted under the crystallization temperature. This result is basically consistent with the theoretical calculation results of chemical reactions and species of slag, as shown in Figure 5a. However, spinel ($MgAl_2O_4$), $CaTiO_3$, and olivine ($[Mg,Fe]_2SiO_4$) were not observed due to their concentrations being below the XRD limit. There was no considerable difference in the crystal structure for the prepared glass-ceramic as the heat-treatment time increased from 30 min to 4 h. This result indicated that these formed glass-ceramics have good thermal stability. Meanwhile, the results of the leaching efficiency of prepared glass-ceramics showed that the leaching efficiency of heavy metal ions was below the limit value, as shown in Table 3. It is feasible to directly prepare glass-ceramics from glass slag via a one-step heat-treatment process. It can be further applied in various fields, such as electronic packing, reinforced glass, and liquid crystal displays, due to its advantages such as a lower thermal expansion coefficient, ease of handling and manufacturing, and lower relative permittivity [44,45].

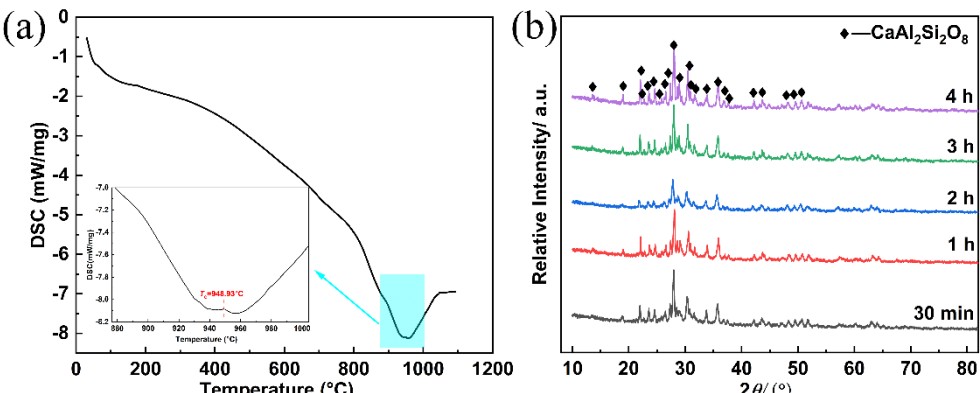

**Figure 8.** The crystallization temperature (**a**) of parent glass obtained from smelting slag after water cooling, and the XRD phases (**b**) of prepared glass-ceramics at different heat-treatment times.

## 4. Conclusions

This work proposed a novel integrated and environmentally friendly capture process to extract PGMs from SACs and efficiently reuse RM simultaneously. In this paper, a mechanism of capture of iron was also studied to better understand the capture process of PGMs. This approach has some advantages, including low cost by avoiding collectors and reducing additive addition, reducing energy consumption by lowering the smelting temperature and shortening the smelting time, and being environmentally friendly by immobilizing heavy metals in glassy slag and glass-ceramics. Several conclusions can be obtained as follows:

1. PGMs in the SAC were trapped by iron from the reduction process of RM in the form of PGM-bearing iron alloys. The optimal smelting conditions were confirmed to be 40–50 wt% of RM additions, 0.8 of basicity, 1500 °C of smelting temperature, and 40 min of holding time, at which the PGM recovery rates were approximately 100%. The PGMs content in obtained slag was less than 1 g/t.

2. A mechanism of the iron collection process, including migration of PGMs during the formation of PGM-bearing alloys and separation of the slag and metal phases, was proposed. The PGMs in molten slag can transfer spontaneously from the slag phase into the metal phase to form the PGM–Fe alloy by thermodynamic analysis, including surface tension and binding energy, and the delocalized electrons theory. Afterwards, the formed PGM—Fe alloy is more readily separated from the slag phase due to the considerable difference density between the slag and metal phases, approximately 4.5 g/cm$^3$, and the low viscous force during the sedimentation process of the metal phase in molten slag.

3. The extracted PGMs from the SAC entered the iron matrix, in which Pd and Rh dispersed in the iron matrix, while Pt was distributed along a certain grain boundary.

4. The synthesis of glass-ceramics was successfully achieved by a one-step heat-treatment process at 950 °C for 30 min to 240 min. Consequently, the main phase of CaAl$_2$Si$_2$O$_8$ was determined in the glass-ceramics.

In summary, this work provides a high-efficiency, economical, and environmentally friendly method for the extraction of PGMs from SACs and simultaneous reutilization of RM.

**Supplementary Materials:** The following are available online at https://www.mdpi.com/article/10.3390/min12030360/s1, Table S1: Relationship between partial molar volume of different oxides and temperature [46–50]; Table S2: The viscosity of slag obtained from CE under different temperature.

**Author Contributions:** C.L.: data curation, investigation, writing–original draft. S.S.: resources, methodology, and funding acquisition. G.T.: conceptualization, supervision. F.X.: writing—review and editing. All authors have read and agreed to the published version of the manuscript.

**Funding:** The work was supported the National Key Research and Development Program of China (2019YFC1907500).

**Data Availability Statement:** The study did not report any data.

**Acknowledgments:** The study was financially supported by the National Key Research and Development Program of China (2019YFC1907500).

**Conflicts of Interest:** The authors declare no conflict of interest.

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
