# Peer review of "An Integrated Capture of Red Mud and One-Step Heat-Treatment Process to Recover Platinum Group Metals and Prepare Glass-Ceramics from Spent Auto-Catalysts"

_minerals, doi:10.3390/min12030360_

Round 1

Reviewer 1 Report

Indicate the software used in for figures 3a and 4c

For Section 3.2.1 present the reactions involved for each phase 

Figure 7 should show the proposed mechanism

Reviewer 2 Report

This manuscript presents a pyrometallurgical process for extraction of PGMs from spent autocatalysts using red mud produced during the Bayer Process as flux and as an iron source for the collection of PGMs. The paper is well presented and structured, the methodology used seems appropriate, and the results are described in detail. The discussion is not restricted to the direct results but also addresses a mechanism for extracting the PGMs and possible economic advantages associated with the proposed route.

Author Response

Thank you for your comment, we have carefully revised our manuscript according your comments. Please see the attachment.

Reviewer 3 Report

The article that I received for review raises an important problem of new method to recover platinum-group metals from spent auto-catalyst and reuse red mud. The title well reflects manuscript content and properly reflects topic of the presented investigation. The manuscript is well written in general. The experimental and discussion parts are understandable and the results make sense. An abstract appropriately indicates the experimental approach and used methods. The introduction is acceptably long and detailed to provide a general outline to a concept of interest. The experimental chapters are well written. Conclusions in summary are clear and adequate. References are sufficient. I appreciate the efforts of the Authors and admire the multitude of the used research methods. In my opinion, the article includes innovative results and could be of interest to readers. I recommend it for publishing.

Author Response

(The authors gave the same response as above.)
